# Prevalence and Risk Factors for Anemia in Non-pregnant Childbearing Women from the Chinese Fifth National Health and Nutrition Survey

**DOI:** 10.3390/ijerph16071290

**Published:** 2019-04-10

**Authors:** Yichun Hu, Min Li, Jinghuan Wu, Rui Wang, Deqian Mao, Jing Chen, Weidong Li, Yanhua Yang, Jianhua Piao, Lichen Yang, Xiaoguang Yang

**Affiliations:** Key Laboratory of Trace Element Nutrition of National Health Commission, National Institute for Nutrition and Health, Chinese Center for Disease Control and Prevention, Beijing 100050, China; huyc@ninh.chinacdc.cn (Y.H.); limin@ninh.chinacdc.cn (M.L.); wujh@ninh.chinacdc.cn (J.W.); wangrui@ninh.chinacdc.cn (R.W.); maodq@ninh.chinacdc.cn (D.M.); chenjing@ninh.chinacdc.cn (J.C.); liwd@ninh.chinacdc.cn (W.L.); yangyh@ninh.chinacdc.cn (Y.Y.); piaojh@163.com (J.P.)

**Keywords:** anemia, hemoglobin, childbearing women, China

## Abstract

Anemia is a public health issue for developing countries, especially for women of childbearing age. The aim of this study was to assess the anemia status and analyze the risk factors for anemia in Chinese childbearing women aged 18–49 years. Hemoglobin concentration was measured by the HiCN method in the Fifth Chinese National Nutrition and Health Survey (CNNHS) in 2010–2012. Age, region type, ethnicity, bodyweight, height, education, season and smoke habit were recorded in unified questionnaires. Latitude was divided by China’s Qinling Mountains and Huaihe River. Childbearing women (28,289) from the CNNHS 2010–2012 were included in this study. The median hemoglobin concentration was 136.2(126.6–145.0) g/L, and it was significantly higher than in CNNHS 2002 (132.5 (122.3–141.6) g/L). The prevalence of anemia was 15.0%, and it was significantly lower than 10 years ago. The logistic regression analysis showed anemia in Chinese childbearing women was specifically related to 30–39 age group (*P* = 0.004), in spring (*P* < 0.0001) or in winter (*P* = 0.006), small and medium-sized cities (*P* = 0.044) and middle school education level (*P* = 0.027). The results showed that anemia status among childbearing women was greatly improved over 10 years since 2002, but it was still more severe than the rest of the populations. The nutrition propaganda and education is recommended for childbearing women to help them to improve the nutritional status on their own.

## 1. Introduction

Anemia is a multifactorial disease that can act both as a risk factor or a consequence of diseases which may affect nervous system, respiratory and circulatory system, skin mucous membrane, digestive system, endocrine system, etc. [1]. The symptoms are nonspecific and are clinically detectable only when the anemia is moderate or severe. It is reported that 42.6% global population (273.2 million people) suffered from anemia in 2011 [2]. 

Anemia is a public health issue for developing countries, especially for children and women [3]. Women of childbearing age are not only experiencing a special period of menstruation, pregnancy, childbirth, lactation, etc., but also play an important role in the economy, society and family. Therefore, their health status deserves special attention. Anemia in childbearing women not only endangers the health of the mother, but also has a bad effect on the next generation once they get pregnant, such as increased risk of preterm delivery and low birth weight [4]. The World Health Organization (WHO) reported that 528 million (29.4%) childbearing women are suffering from anemia around the world [5,6]. Chinese residents had a prevalence of anemia at 20.1% according to the fourth Chinese National Nutrition and Health Survey (CNNHS 2002), and the anemia prevalence for childbearing women was 26.1% [7]. In the fifth round of CNNHS, the hemoglobin (Hb) concentration was also measured during 2010–2012. 

The objective of this study is to evaluate the anemia status of Chinese childbearing women in CNNHS2010–2012 by detecting whole blood Hb concentration, and to analyze the risk factors of anemia including population informatics, geographic information, etc. The changes in anemia status during the 10 years were also analyzed.

## 2. Materials and Methods 

### 2.1. Subjects and Ethics

The CNNHS is a cross-sectional survey of the civilian non-hospitalized population of China [8], conducted by the National Institute for Nutrition and Health, Chinese Center for Disease Control and Prevention (NINH, China CDC). All the participants of this survey were selected by using a stratified proportional random cluster sampling. The specific sampling method was elaborate by Hu et al. [9]. A total of 28,289 childbearing women aged 18–49 years (non-pregnant) were included in this study. Informed consents were obtained from all the participants in writing to participate in the study. Ethics Review Board of Institute for Nutrition and Health, China CDC approved the protocol (2013–2018). 

### 2.2. Blood Samples and Hemoglobin Measurement

A national project workgroup was established in NINH, China CDC to develop a unified survey and questionnaire, and to carry out the investigation survey by using unified tools and methods. Hb concentration was measured in the CDC laboratory in the district (county) of the survey areas. Laboratories operation staffs participated in the uniformed training and examination. After becoming qualified, they took part in the Hb measurement.

Fasting venous blood was first collected in ethylenediamine tetraacetic acid (EDTA) tubes, and then 10 µL anticoagulant whole blood was collected using a quantitative capillary tube (Drummond Scientific Company, Broomall, PA, USA) for Hb determination. The HiCN method recommended by International Committee for Standardization of Hematology [10] is the most reliable laboratory method for the quantitative determination of Hb, and serves as a reference for the comparison and standardization of other methods [1]. The quality control and blind samples were made by Beijing XiBai Biomedical, Inc. (Beijing, China) under the unified commission of the National Working Group of National Institute for Nutrition and Health, China CDC, and then distributed to all the survey areas by cold chain transportation. Blind samples (high and low value Hb) were measured and qualified before proceeding with fieldwork. After the formal start of the fieldwork, a series of quality control analysis (including quality control samples’ analysis and blind samples’ analysis) was performed before sample testing and then performed every thirty samples. Each sample required dual sample determination. The qualified rate of quality control samples was 90%. The intra- and inter-assay CV (coefficient of variation) in the analysis method were 1.03–2.91% and 2.04–3.70%, respectively.

### 2.3. Criteria of Anemia

The Hb concentration varies with age, gender, pregnancy and altitude. The cut-off for non-pregnant childbearing women formulated by WHO is 120 g/L [11]. The anemia criteria should be adjusted according to the altitude for those who lives in areas above 1000 m altitude, as suggested by WHO in 2001 [1]. The severity of anemia was classified according to the Hb concentration recommended by WHO, as to non-pregnant women, 110 g/L ≤ Hb < 120 g/L mild anemia; 80 g/L ≤ Hb < 120 g/L, moderate anemia; Hb < 80 g/L, severe anemia [11]. 

### 2.4. Variables

All the information of participants in this study was collected from every survey area and logged into the unified software “China National Nutrition and Health Survey System Platform”. Region type and latitude were recorded and classified [8]. Based on self-reporting, demographic data (including age, ethnicity, education, smoke habit, annual family income, etc.) was recorded. The altitude was recorded by the local investigators. BMI was calculated according to bodyweight and height. Body weight was measured by using Double lever weight scale, and standing height was measured by metal column type height meter. Season was recorded according to the month of blood collection (March to May, spring; June to August, summer; September to November, autumn; December to February, winter). China’s Qinling Mountains and the Huaihe River were recognized as the boundary to divide the North and the South. 

### 2.5. Data Check and Analyses

The data of Hb concentration of childbearing women from each investigation area was input into the unified survey platform. According to the unified standard of data cleaning principle (Appendix A), the problematic record was then returned to each area for re-inspection. All the batches of samples with unqualified quality control samples were eliminated to ensure the accuracy of the results. The data of Hb concentration of childbearing women in 2002 was obtained from CNNHS 2002 database. 

All the data collected was analyzed by SAS 9.4 software (SAS Institute Inc., Cary, NC, USA). We utilized univariate (proc univariate) and frequency procedure (proc surveyfreq) which had weighted percentage and 95% confidence intervals (CIs). All the participants in this study were divided into different sub-groups according to different hypothesized predictors for anemia status. The hemoglobin concentration did not conform to normal distribution, therefore the blood Hb concentrations were then compared by Kruskal-wallis test for multiple comparisons and DSCF (Dwass, Stel, Critchlow-Fligner) test for comparisons between two subgroups. Frequencies were presented as percentages (%) and the rates of subgroups were compared by Rao-scott test, and the P value was corrected by Bonferroni method while comparing the difference between subgroups. Multivariable logistic regression analysis (proc surveylogistic) was utilized to analyze the relationship between anemia and possible predictors (e.g., age group, region type, ethnicity, latitude, season, BMI level, education, smoke habit and annual family income). The odds ratio (OR) and 95% confidence intervals (CIs) were determined by multivariable logistic regression models. The difference was statistically significant with *P* < 0.05. 

## 3. Results

### 3.1. Participant Characteristics

The Hb concentration of 28,289 healthy female aged 18–49 years was included covering 30 provincial administrative regions (Figure 1) were analyzed in this study (Table 1). Age groups were classified according to three different stages, and only 19.0% participants aged below 30 years old. The survey participants came from four different region types according to the economic development level of each survey area. 86.9% were from the Han ethnicity and only 13.1% was from the minorities. The survey covered four seasons of the years, but mainly in the autumn (68.3%). 60.6% participants had normal BMI and 1.8% smoked every day. 11.7% the participants had college education and above. 48.2% had annual family incomes lower than 10,000 yuan.

### 3.2. Blood Hemoglobin of Chinese Childbearing Women

The concentration of blood Hb for all participants of different sub-groups in CNNHS 2010–2012 was presented in Table 1. The median blood Hb was 136.2 g/L. Statistical differences were found in different age groups, region types, latitude, ethnicities, latitude, seasons, BMI levels, education levels, smoke habits and annual family incomes in terms of Hb concentration. The median concentration of Hb from small and medium-sized cities was significantly lower than the other region types, while the median Hb concentration from the poor areas was significantly higher than the other areas. Han ethnicity had lower Hb concentration than other ethnicities. Participants from the south areas had lower Hb concentration. In spring, the level of Hb was significantly lower than the other seasons. In particular, participants who smoke daily had a significantly higher median Hb concentration (139.7 g/L) than the other groups.

### 3.3. Anemia Prevalence of Chinese Childbearing Women

The anemia prevalence of Chinese childbearing women in 2010–2012 was 15.0% (Table 1). Childbearing women who had a habit of smoking on a daily basis had the lowest anemia prevalence (10.8%). The youngest childbearing women had the lowest anemia among all age groups. The prevalence of anemia varied among seasons. In spring, it showed that the prevalence of anemia was above 30%, which was significantly higher than all the other seasons. Participants whose education level were college and above had significantly lower anemia prevalence (11.8%) than other education groups. The anemia prevalence was observed gradually decreasing by the increased annual family income except the 30,000–39,999 yuan group with wide CI. Although significant difference was found in different region types in terms of median Hb concentration (*P* < 0.0001), no significant difference was found in terms of anemia prevalence (*P* = 0.058). No significant difference was found in different ethnicity, BMI level, and latitude in terms of anemia prevalence.

### 3.4. Risk Factors for Anemia in Chinese Childbearing Women

In multi-variable logistic regression analysis for all the participants, we analyzed the factors including age group, region type, ethnicity group, latitude, season, BMI level, education level and annual family income (Table 2). The results of logistic regression analysis showed, the anemia was associated with aged 30–39 years (OR = 1.260, *P* = 0.004; comparing to those aged 18–29 years), small and medium-sized cities (OR = 1.167, *P* = 0.004; comparing to those from large cities), middle school education (OR = 1.300, *P* = 0.027; comparing to those had college education and above). Taking autumn as the reference standard, spring (OR = 2.659, *P* < 0.0001) and winter (OR = 1.258, *P* = 0.006) increase the risk of anemia. Besides, it showed that participants who did not smoke or occasionally smoke had increased the risk of anemia (OR = 1.484, *P* = 0.041).

### 3.5. Comparison of anemia status over ten years

We compared the Hb concentration and the anemia prevalence in the CNNHS 2010–2012 with that in CNNHS 2002. There were 44,224 childbearing women aged 18-49 years included in the CNNHS 2002, and 28,289 were included in CNNHS 2010–2012. The median Hb concentration in 2010–2012 was 136.2(126.6–145.0) g/L, and it was 132.5(122.3–141.6) g/L in 2002. After 10 years of nutritional improvements, the median Hb concentration in each age group, region and ethnicity in CNNHS 2010–2012 was significantly increased (*P* < 0.001, Figure 2). 

The anemia prevalence of childbearing women aged 18–49 years in 2002 was 26.0% and it decreased significantly in 2010–2012 by 42.3%. The anemia prevalence in CNNHS2010–2012 was significantly decreased comparing to CNNHS2002 in different age groups and region types. Participants aged 18–29 years had the best anemia improvement, and the anemia rate decreased by 44.9% than that of CNNHS2002, followed by the 40–49 age group and 30–39 age group (Figure 3). Childbearing women from rural areas had better anemia improvement at 46.5% reduction than those from cities. Among childbearing women with anemia, there was 58.8% had mild anemia, 38.9% had moderate anemia and 2.32% had severe anemia in CNNHS2002, while there was 52.1% had mild anemia, 43.9% had moderate anemia and 4.0% had severe anemia in CNNHS2010–2012. Though the anemia prevalence decreased over the past 10 years, the proportion of moderate to severe anemia increased (Figure 4).

## 4. Discussion

In CNNHS 2010–2012, the median Hb concentration of Chinese childbearing women aged 18–49 years was 136.2 g/L (mean Hb concentration was 135.0 g/L) which was considerably higher than that of CNNHS 2002 (132.5 g/L, mean Hb concentration was 131.4 g/L). The Hb concentration was also significantly higher than the global mean Hb 126.0 g/L in non-pregnant women aged 15–49 years in 2011 [12]. From CNNHS 2002 to CNNHS 2010–2012, a significant decrease in the prevalence of anemia was observed in childbearing women (26.0% to 15.0%) (*P* < 0.001). The anemia prevalence of Chinese childbearing women (15.0%, CI: 14.3–15.7%) was significantly lower than the global prevalence of anemia in 2011 (29.4%, 95% CI: 24.5–35.0) [2]. The anemia prevalence of Chinese childbearing women was close to women aged 18–49 years from a neighboring country like Korea (14.7%) [13]. However, it was still higher than childbearing women from Canada (4%, Canadian Health Measures Survey, 2009 to 2011) [14] and non-Hispanic White American (3.6~6.6%) [15]. Anemia in Chinese childbearing women would be classified as a mild public health problem, while it was classified as moderate 10 years ago according to the WHO proposed classification based on prevalence [11]. This is a great improvement, which should be attributed to the nutrition improvement programs over the ten years. In the year of 2001, the general office of the State Council issued the Outline of China’s Food and Nutrition Development (2001–2010) [16]. The Outline clearly pointed out that we should focus on the development of fortified food. Started at 1995, the National Public Nutrition Improvement Action promoted the "National ten year investment plan (2003–2012)" to improve public nutrition in a variety of forms, such as dietary nutrition, food fortification, and nutritional supplements [17]. In 2002, the State Administration of Grain and the Ministry of health, started a pilot plant for nutrition strengthening. Fortified micronutrients, including iron, zinc, calcium, vitamin B1, vitamin B2, folic acid, nicotinic acid, were strengthened in wheat flour. Subsequently, China implemented ferric fortified soy sauce program to prevent and improve anemia since 2003. In addition, remarkable progress had been also made in health care reform over the past ten years, such as the rapid development of community health service, the expansion of the new rural cooperative medical system and so on. The implementation of these policies and programs had effectively improved the nutritional status of Chinese residents. Based on the successful experience of China over the 10 years. Though the anemia prevalence had dropped significantly, however, the prevalence of anemia among women of childbearing age was still high (15.0%). It was even higher than that of children (5.0%) and the elderly (12.6%) [8]. Childbearing women aged 18–49 years are the backbones of work and family, thus it is still worth paying great attention to the problem of anemia in Chinese women of childbearing age. Childbearing women aged 30–39 years had the highest anemia rate among all the age groups, and the same trend was also found in 2002 [7]. This might because women of this age were at the craziest time toward family and work and most of them already had child, which meant they did not have enough time and energy to pay attention to the nutritional status of their own. 

The Hb concentration of childbearing women from poor rural areas was significantly higher than that from the other areas. High proportion of participants from high altitude area, 33.3% (1941/5825), might explain this phenomenon. As the body compensates for hypoxic hypoxia at high altitude, the Hb concentration of body will increase. According to the recommendation of WHO, the criteria for anemia will rise as the elevation increases, which explained the increase in Hb concentration in poor rural areas, and the anemia prevalence did not decrease. The median Hb concentration of the small and medium-sized cities was the lowest. The anemia prevalence of small and medium-sized cities was significantly higher than that of large cities, but not significantly different from that of rural areas. Similar trend was also found in CNNHS2002 [7]. Most of the small and medium-sized cities in China were the combination of urban and rural areas with complex source of population. The change of life style, diet habits, and etc might be factors to affect the anemia situation, and further research was needed on the underlying causes. 

The Hb concentration was lowest in spring. Altitude may be one of the reasons. There are no samples from high altitude areas in spring, while all the other seasons contained certain quantities of samples from high altitude areas (summer, 125/3098; autumn, 1677/19314; winter, 139/4641). High proportion samples from small and medium-sized cities might be another reason. 84.6% participants were from small and medium-sized cities, which was far more than the proportion of participants from the same region type in the other seasons. Moreover, we think the custom of replenishing in autumn for Chinese people might partly explain the lowest anemia prevalence in autumn. The seasonal factor on anemia was rarely reported. Ronnenberg et al. reported that the Anqing women of Childbearing Age showed twice the moderate anemia prevalence in summer than those in winter [18] because of seasonal variation in B vitamin level status. Seasonal factors might be an important factor affecting anemia regarding food diversity, therefore, further study are required to analyze the specific reasons.

With the education level increased, the prevalence of anemia was gradually reduced. This might because people with higher level of education had stronger awareness of improving nutritional status of their own. Similar trend was also found in terms of annual family income. People with higher incomes tend to have more awareness and ability to improve their own nutrition. 

Participants with daily smoking habita had significantly higher Hb concentration and lower anemia prevalence than non-smokers or seldom smokers. In the 1970s, scientists confirmed that smoking could increase Hb concentration. The carbon monoxide bonds to Hb to form carbonhemoglobin (HbCO), an inactive Hb type without oxygen carrying capacity [19]. Mast et al. reported that cigarette smoking increased Hb by 2.6–5.9 g/L depending on the intensity [20]. Nordenberg et al. reported that the mean Hb concentration was significantly higher for those who smoked 10 or more cigarettes per day and they suggested adjusting the cutoff of anemia screening for smokers [19]. In our study, the median Hb concentration of smokers was higher than non-smokers or seldom smokers by 3.5 g/L. The results were in conformity with previous reports. The increase of Hb concentration caused by smoking might mask their anemia condition, especially for childbearing women. However, we can hardly make a recommendation of adjustment value because our study was not a special-designed controlled trial to study the effect of smoking on the judgment threshold of anemia.

We noted several shortcomings of this study. Firstly, a limited number of participants or anemic individuals in several subgroups (such as age groups, season, smoke habit and annual family incomes) might affect the estimate as a weighted proportion. Therefore, estimates with small numbers of anemic subjects or wide CIs should be carefully interpreted. Secondly, there are many reasons affecting the anemia status for childbearing women, for example, iron deficiency, G6PD deficiency, thalassemia, lack of vitamin B12, menstrual cycles, menstrual blood volume and etc. Among them, iron deficiency is usually the main reason. The characteristics of Chinese residents’ diets are mainly plant foods, therefore iron deficiency is usually the main cause for anemia because of plant-derived non-heme iron with low absorption rate. Therefore, iron-deficient anemia may be the main reason for Chinese anemia population [6]. Besides, thalassemia and anemia caused by G6PD deficiency might also be the reasons for Chinese residents, especially for those from Guangdong, Guangxi, Yunnan, Guizhou, and Hainan provinces [21,22]. However, the types of anemia status were not included in CNNHS. We can hardly further speculate the specific reasons of anemia for childbearing women in China. 

## 5. Conclusions

In general, the anemia prevalence of Chinese childbearing women in 2010–2012 was significantly improved over the 10 years. However, it was still higher than the total average anemia prevalence of Chinese population. Reducing anemia is recognized as an important component of the health improvement of women and children. The second global nutrition target for 2025 calls for a 50% reduction of anemia in women of reproductive age [2]. In the year of 2014, the general office of State Council issued the Outlines of China’s Food and Nutrition Development (2014–2020) [23]. The newly-issued outlines proposed to strengthen nutrition improvement and health education, increase the nutrition monitoring and intervention, and women was one of the key target population group for nutrition improvement. According to the results of this study, the nutrition propaganda and education is recommended continuously in all region types, especially in small and medium-sized cities and those with lower education levels. We encourage women of childbearing age to pay more attention to the nutritional health of their own, especially for those during 30–39 years old. 

## Figures and Tables

**Figure 1 ijerph-16-01290-f001:**
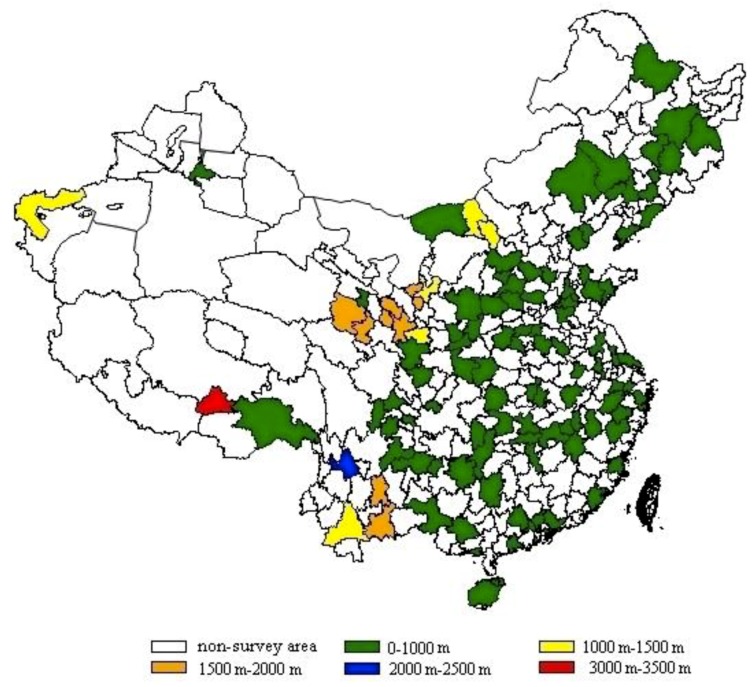
Survey areas distribution and its altitude (the municipal administrative units of survey area were showed. the Hong Kong Special Administrative region, Macao Special Administrative Region and Taiwan province were not included in this study. There was no survey area at altitude between 2500 and 3000 m).

**Figure 2 ijerph-16-01290-f002:**
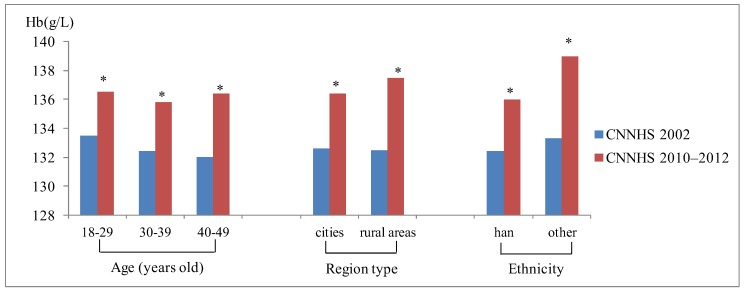
Blood hemoglobin concentration of the fourth Chinese national nutrition and health survey (CNNHS 2002) and the fifth CNNHS 2010–2012, stratified by demographic characteristics. (*significant differences between CNNHS 2010 and 2002, *P* < 0.001).

**Figure 3 ijerph-16-01290-f003:**
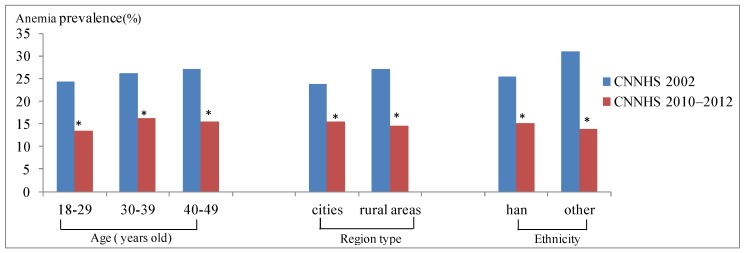
Prevalence of anemia for the fourth Chinese national nutrition and health survey (CNNHS 2002) and the fifth CNNHS 2010–2012, stratified by demographic characteristics. (*significant differences between CNNHS 2010 and 2002, *P* < 0.001).

**Figure 4 ijerph-16-01290-f004:**
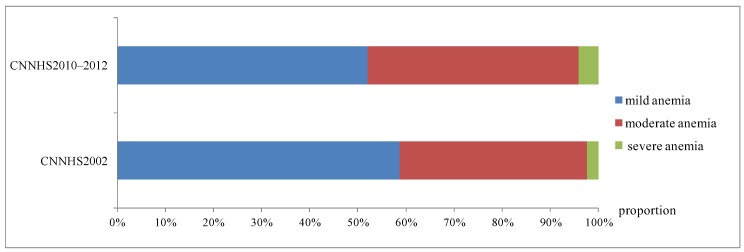
Distribution of anemia in different severity in the fourth Chinese national nutrition and health survey (CNNHS 2002) and in CNNHS 2010–2012 (mild, moderate and severe refer to different degree of anemia; Proportion indicates the proportion of anemia to total anemia in different degrees.).

**Table 1 ijerph-16-01290-t001:** Hemoglobin concentration and anemia prevalence for childbearing women from the China National Nutrition and Health Survey 2010–2012.

Variables	*N*(%)	Weighted Hemoglobin (g/L)^1^	*P* ^2^	Weighted prevalence (%, 95%CI)	*P* ^3^
Total	28,289	136.2(126.6–145.0)		15.0(14.3–15.7)	
Age group			<0.0001 χ^2^ = 19.21		0.001 F = 6.62
18–29 years old	5370(19.0)	136.5(127.4–145.2)^a^		13.4(12.0–14.8)^b^	
30–39 years old	9034(31.9)	135.8(126.0–144.2)^b^		16.3(15.2–17.4)^a^	
40–49 years old	13,885(49.1)	136.4(126.4–145.3)^a^		15.5(14.6–16.4)^a^	
Region type			<0.0001 χ^2^ = 125.29		0.058 F = 2.49
Large cities	5607(19.8)	137.6(127.4–145.6)^a^		13.3(11.8–14.7)^b^	
Small and medium-sized cities	7962(28.1)	134.7(125.6–142.9)^b^		15.8(14.6–17.0)^a^	
Ordinary rural areas	8895(31.5)	137.4(127.7–146.4)^a^		14.3(13.3–15.4)^a^	
Poor rural areas	5825(20.6)	138.5(128.5–148.5)^c^		14.9(13.4–16.4)^a^	
Ethnicity			<0.0001 χ^2^ = 116.96		0.208 F = 1.59
Han	24,596(86.9)	136.0(126.3–144.6)		15.1(14.4–15.8)	
Other	3693(13.1)	139.0(129.4–148.1)		13.8(12.1–15.6)	
Latitude			<0.0001 χ^2^ = 162.18		0.057 F = 3.62
North	12,872(45.5)	136.4(126.3–145.9)		15.8(14.6–16.9)	
South	15,417(54.5)	136.0(126.8–144.3)		14.4(13.6–15.2)	
Season			<0.0001 χ^2^ = 56.03		<0.0001 F = 29.40
Spring	1236(4.4)	130.9(116.7–142.4)^a^		30.3(25.4–35.2)^a^	
Summer	3098(11.0)	135.2(125.0–144.4)^b^		17.9(15.7–20.2)^b^	
Autumn	19,314(68.3)	136.8(127.6–145.4)^c^		13.5(12.7–14.3)^c^	
Winter	4641(16.4)	135.0(125.5–143.5)^b^		16.5(14.7–18.3)^b^	
BMI^4^ level			<0.0001 χ^2^=108.37		0.957 F = 0.04
BMI < 24	17161(60.6)	135.8(126.4–144.4)^a^		15.0(14.1–15.9)	
24 ≤ BMI < 28	8196(29.0)	137.0(126.9–146.0)^b^		14.8(13.6–16.0)	
BMI ≥ 28	2932(10.4)	137.0(127.6–146.9)^b^		15.2(13.0–17.3)	
Education			<0.0001 χ^2^=49.63		0.002 F = 6.49
Primary school	8709(30.8)	136.6(126.8–146.2)^a^		15.7(14.5–16.8)^a^	
Middle school	16,280(57.5)	136.1(126.3–144.8)^b^		15.3(14.4–16.2)^a^	
College and above	3300(11.7)	136.2(127.6–143.7)^b^		11.8(10.0–13.5)^b^	
Smoke habits			<0.0001 χ^2^=19.63		0.048 F = 3.91
Smoke daily	513(1.8)	139.7(129.0–150.4)		10.8(7.2–14.4)	
No or occasionally	27,776(98.2)	136.2(126.5–145.0)		15.0(14.4–15.7)	
Annual family income (yuan)			<0.0001 χ^2^=34.91		0.005 F = 3.34
<10000	13,637(48.2)	136.2(126.3–145.3)^b^		16.2(15.1–17.2)^a^	
10,000–19,999	8156(28.8)	136.5(126.9–145.0)^a^		13.6(12.5–14.8)^b^	
20,000–29,999	2770(9.8)	136.1(127.8–143.8)^ac^		13.1(11.1–15.1)^b^	
30,000–39,999	978(3.5)	134.2(125.2–142.2)^ac^		16.2(12.5–19.8)^ab^	
40,000–	1026(3.6)	136.5(127.0–144.4)^a^		13.0(10.0–16.0)^ab^	
Not answer	1722(6.1)	136.8(126.0–145.0) ^4^		15.0(12.2–17.7)^4^	

*^1.^* P50 (P25–P75), all such values; ^2^ P value for the Kruskal–Wallis test. ^3^ P value for theRao-scott testt; ^4^ BMI-Body mass index; *^4^* Not included in subgroups’ comparison. ^abc^ significant differences between subgroups.

**Table 2 ijerph-16-01290-t002:** Determinants of anemia among Chinese childbearing women from Chinese National Nutrition and Health Survey 2010–2012 (logistic regression).

Variables	Total (*n* = 14473) odds ratio	*P*
Age group		
18–29 years old	ref	
30–39 years old	1.260(1.087–1.461)	0.004
40–49 years old	1.459(1.002–1.339)	0.552
Region type		
Large cities	ref	
Small and medium-sized cities	1.167(0.999–1.364)	0.044
Ordinary rural areas	1.003(0.848–1.187)	0.157
Poor rural areas	1.125(0.917–1.380)	0.417
Ethnicity		
Other	ref	
Han	1.148(0.967–1.362)	0.115
Latitude		
South	ref	
North	1.043(0.930–1.169)	0.474
Season		
Autumn	ref	
Spring	2.659(2.070–3.417)	<0.0001
Summer	1.478(1.229–1.779)	0.907
Winter	1.258(1.090–1.452)	0.006
BMI level		
BMI < 24	ref	
24 ≤ BMI < 28	0.925(0.821–1.042)	0.514
BMI ≥ 28	0.934(0.821–1.042)	0.747
Education		
College and above	ref	
Primary school	1.286(1.027–1.610)	0.117
Middle school	1.300(1.065–1.585)	0.027
Annual family income (yuan)		
40,000–	ref	
<10,000	1.173(0.881–1.562)	0.077
10,000–19,999	0.958(0.718–1.278)	0.078
20,000–29,999	0.945(0.687–1.301)	0.160
30,000–39,999	1.241(0.851–1.811)	0.188
Smoke habits		
Smoke daily	ref	
No or occasionally	1.484(1.016–2.167)	0.041

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
