# Peer review of "Prevalence and Risk Factors for Anemia in Non-pregnant Childbearing Women from the Chinese Fifth National Health and Nutrition Survey"

_ijerph, 2019, doi:10.3390/ijerph16071290_

Round 1
Reviewer 1 Report
This manuscript investigates the prevalence of anemia and the factors that predict anemia among non-pregnant childbearing age women in China. While anemia is a very serious issue due its global prevalence and adverse effect especially among women of child bearing age, there are some concern that need to be fully addressed.
The manuscript contains several grammatical errors that affects the presentation of information in this study. This runs through most sections of the manuscript.
Line 108: Authors used a nonparametric test to compare hemoglobin values. Since most of the times, hemoglobin concentration is normally distributed, the authors should give a justification for using the non-parametric test.
There is no mention of a post-hoc test after conducting the Kruskal-Wallis test. It is not clear by what means the authors are able to tell which subgroups are significantly different from others.
Line 145-146: Anemia prevalence did not differ by region type according to table 1, contrary to what is depicted by the authors. This mistake is also repeated in the discussion (Line 238-240). Though a case for significantly higher anemia prevalence can be made from the regression analysis using the odds ratio, it can only be compared to the reference group.
Line 152: Not clear what authors mean by “prevalence.3.4 Risk factors”
Table 2: “women” omitted from table title
Line 199: Is this global mean for only women of child bearing age or for all ages?
Author Response
Point 1: Line 108: Authors used a nonparametric test to compare hemoglobin values. Since most of the times, hemoglobin concentration is normally distributed, the authors should give a justification for using the non-parametric test.
Response to Point 1:In the first step of data analysis, we did normal tests on the data by using proc univariate procedure (SAS 9.4). The result showed that the concentration of hemoglobin did not conform to normal distribution. Therefore we utilized the non-parametric test method to compare the hemoglobin concentration of different groups. We added explanation in the Part 2.5 Data check and analyses (Line 114).
Point 2:There is no mention of a post-hoc test after conducting the Kruskal-Wallis test. It is not clear by what means the authors are able to tell which subgroups are significantly different from others.
Response to Point 2: The description of statistical method was added in the revised manuscript in line 118. We adopted the method Bonferroni to perform comparison between subgroups.
Point 3: Line 145-146: Anemia prevalence did not differ by region type according to table 1, contrary to what is depicted by the authors. This mistake is also repeated in the discussion (Line 238-240). Though a case for significantly higher anemia prevalence can be made from the regression analysis using the odds ratio, it can only be compared to the reference group.
Response to Point 3: In the last sentence of this paragraph, we clarify that there was no difference in anemia prevalence among different regions. In line 163-164, we meant to express that small and medium-sized cities had relatively high anemia prevalence and large cities had relatively low anemia rates. However, the expression was not rigorous enough. We deleted the sentence in the revision version.
Point 4: Line 152: Not clear what authors mean by “prevalence.3.4 Risk factors”
Response to Point 4: We are so sorry about this type error. A return key should be added before ‘3.4 Risk factors’.
Point 5: Table 2: “women” omitted from table title.
Response to Point 5: We corrected the mistake.
Point 6: Line 199: Is this global mean for only women of child bearing age or for all ages?
Response to Point 6: The global mean was for non-pregnant women aged 15–49 years in 2011.
Reviewer 2 Report
This is a comprehensive report of the results of the fifth Chinese national nutrition and health survey (CNNHS) in 2010-2012.
Despite worldwide economic and scientific development, more than a quarter of the world’s population remains anemic, and about half of this burden is a result of iron deficiency anemia (IDA). IDA is most prevalent among preschool children and women. Among women, physical and cognitive performance, work productivity, and well-being require adequate iron supply; also iron during pregnancy improves maternal, neonatal, infant, and even long-term child outcomes.
Public health interventions to reduce the burden of IDA are complex and required well established programs. The results of this fifth Chinese national nutrition and health survey 2010-2012 show the prevalence of anemia in this period is significantly improved compared with that of 10 years ago, clearly reflect the success of China in the improvement of the women health.
Results are well presented and the potential weak points addressed
General comments
The main concern is the Hb measurement.
A variety of methods for measuring hemoglobin are available in the laboratory, at point-of-care, and in the field. Although the cyanmethemoglobin method is routinely used in automated instruments, in many low-income settings, photoelectric calorimeters can perform manual cyanmethemoglobin-based measurements; this requires careful dilution of patient samples, regular calibration, and electricity.
Authors report only the intra- and inter-assay coefficient of variation, while total error must be also reported
According to Westgard webpage for hemoglobin the desirable specification for allowable total error is 4.19 % for Hb.
Page 2 lines 76-80
“Blind samples (high and low value Hb) were measured and qualified before proceeding with fieldwork. “
These samples are then used as internal quality control , how were prepared? Is the Disease Control Program responsible for preparing and sending these samples to the different centers ? Please explain, because in western countries commercial quality controls are available, don´t know how it is organized in China
Samples were analyzed in duplicates, who performed the analysis, nurses, medical laboratory professionals?
Second point :altitude
Abstract: “ Age, region type, ethnicity, bodyweight, height, education, season and smoke habit were recorded”
Along the manuscript authors state that
“Based on self-report, demographic data (including age, ethnicity, latitude, education, smoke habit, annual family income and etc”
Also altitude ? Please clarify, becaue in page 2 lines 83-85
“The anemia criteria should be adjusted according to the altitude for those who lives in areas above 1000m altitude, suggested by WHO in 2001”
China has great physical diversity, with mean elevation 1,840 m and high % of inhabitants living in this altitudes.
A map could aid to clarify points and understand the diverse areas , and to locate the cities where the Center for Disease Control are located
For better understanding the results
China is diverse with rural areas and Special economic zones; may be the life style of the citizens are also different enough to consider this point: more traditional in rural villages and western-like for young persons in main cities.
Not only life style, also the diet, the change from the Chinese traditional dishes, healthy and low fat content, to health threaten fast food rich in animal proteins and saturated fats, typical of western way of life.
Season is also an important point to consider when evaluating anemia and iron metabolism; it is well known the effect of season in serum iron and transferrin , and also the diet influences, changing with the climate in winter and summer ie.
Author Response
Point 1: The main concern is the Hb measurement.
A variety of methods for measuring hemoglobin are available in the laboratory, at point-of-care, and in the field. Although the cyanmethemoglobin method is routinely used in automated instruments, in many low-income settings, photoelectric calorimeters can perform manual cyanmethemoglobin-based measurements; this requires careful dilution of patient samples, regular calibration, and electricity.
Authors report only the intra- and inter-assay coefficient of variation, while total error must be also reported
According to Westgard webpage for hemoglobin the desirable specification for allowable total error is 4.19 % for Hb.
Response to Point 1: All the Hb measurements were performed in each survey areas by HiCN method. First of all, the HiCN method is an reliable method for Hb measurements. Secondly, this method is suitable for different survey areas though pre-project investigation and the results are consistent. Third, it is an economical method. All the laboratory technicians were trained and accessed. After becoming qualified, they could take part in the Hb measurement. The quality control samples were added in every 30 samples. The total qualified rate of quality control is 90% (added in the paper). We eliminated all the batches of samples with unqualified quality control samples to ensure the accuracy of the results. In the whole process of data analysis, 2236 unqualified samples were deleted, of which about 400 unqualified samples were women of childbearing age.
Point 2: Page 2 lines 76-80
“Blind samples (high and low value Hb) were measured and qualified before proceeding with fieldwork. “
These samples are then used as internal quality control, how were prepared? Is the Disease Control Program responsible for preparing and sending these samples to the different centers? Please explain, because in western countries commercial quality controls are available, don´t know how it is organized in China.
Response to Point 2: The quality control and blind samples were made by Beijing XiBai Biomedical, Inc. under the unified commission of the National Working Group of National Institute for Nutrition and Health, China CDC, and then distributed to all the survey areas by cold chain transportation. We added the description in line 75-76 in the revised manuscript.
Point 3: Samples were analyzed in duplicates, who performed the analysis, nurses, medical laboratory professionals?
Response to Point 3: The Hb was measured by laboratory technicians in the CDC laboratory of city level for each survey area. All the test results were then input into the ‘China National Nutrition and Health Survey System Platform’. The professional data analysts from National Project Workgroup in National Institute for Nutrition and Health, China CDC performed the analysis according to the principles of data cleaning (Appendix A, Line 326-339).
Second point :altitude
Point 4: Abstract: “ Age, region type, ethnicity, bodyweight, height, education, season and smoke habit were recorded”
Along the manuscript authors state that “Based on self-report, demographic data (including age, ethnicity, latitude, education, smoke habit, annual family income and etc”
Also altitude ? Please clarify, because in page 2 lines 83-85
“The anemia criteria should be adjusted according to the altitude for those who lives in areas above 1000m altitude, suggested by WHO in 2001”
Response to Point 4: The altitude was recorded by the local investigators. We added the description in line 97-98.
Point 5: China has great physical diversity, with mean elevation 1,840 m and high % of inhabitants living in this altitudes.
A map could aid to clarify points and understand the diverse areas, and to locate the cities where the Center for Disease Control are located
Response to Point 5: The national nutrition survey covered 30 provincial administrative regions, including 23 provinces, 5 autonomous regions and 4 municipalities directly under the Central Government, while the Hong Kong Special Administrative region, Macao Special Administrative Region and Taiwan province were not included.
We adjusted the calibrated criteria for anemia according to the altitude of each survey area. We have drawn a map with all survey areas and marked those with an altitude that is more than 1000m by different colors based on your suggestion. We didn't marked the cities where the Center for Disease Control (CDC) are located because all the Hb measurements were carried in the CDC of municipal administrative regions. All the municipal administrative survey regions were showed in our map.
Point 6: For better understanding the results
China is diverse with rural areas and Special economic zones; may be the life style of the citizens are also different enough to consider this point: more traditional in rural villages and western-like for young persons in main cities.
Not only life style, also the diet, the change from the Chinese traditional dishes, healthy and low fat content, to health threaten fast food rich in animal proteins and saturated fats, typical of western way of life.
Response to Point 6: We added the related discussion in the 2nd paragraph of the discussion part. The dilemma was that there was significant difference in Hb concentration but no significant difference in anemia prevalence between different region types. Therefore, we didn't make in-depth discussion about it.
Point 7: Season is also an important point to consider when evaluating anemia and iron metabolism; it is well known the effect of season in serum iron and transferrin, and also the diet influences, changing with the climate in winter and summer ie.
Response to Point 7: In this paper, we found that the season was an important point for anemia situation for childbearing women. The anemia prevalence was lowest in autumn and highest in spring. However, due to the shortcomings mentioned in the article, it is difficult for us to make further analysis based on the existing data.
Round 2
Reviewer 1 Report
This manuscript investigates the prevalence of anemia and the factors that predict anemia among non-pregnant childbearing age women in China. The manuscript has been improved compared to the initial submission. I have put forward a few suggestions to improve the manuscript.
Line 62: "Written informed consent was obtained from" subjects rather than "given to" subjects.
Line 156-157. The table shows a p-value of 0.058 and so the authors must mention that the slightly higher anemia prevalence was not statistically significant otherwise it becomes blurry as to which observations are significant and which ones are due to chance.
Line 193: I am assuming that the rates given in these lines (58.8%, 41.2%, 52.1% etc.) refer to the percentage among people with anemia. It is misleading as it is currently. The current writing suggest that almost half of the childbearing women had mild anemia and the remaining half had moderate to sever anemia.
Table 1: Authors should indicate which groups were significantly different from each other based on post-hoc test results. For example, using superscripts, with groups with significantly different Hb concentration or weighted anemia prevalence values having different superscripts.
There is still need to improve the grammar and language usage in several parts of the manuscript. For example lines 304.
Also using “hemoglobin concentration” is to be preferred over “hemoglobin levels” as levels may also imply categories of hemoglobin concentration.
Author Response
Point 1: Line 62: "Written informed consent was obtained from" subjects rather than "given to" subjects.
Response 1: We have revised it in the manuscript.
Point 2: Line 156-157. The table shows a p-value of 0.058 and so the authors must mention that the slightly higher anemia prevalence was not statistically significant otherwise it becomes blurry as to which observations are significant and which ones are due to chance.
Response 2: We have revised all the related expression in both the result part and the discussion part.
Point 3: Line 193: I am assuming that the rates given in these lines (58.8%, 41.2%, 52.1% etc.) refer to the percentage among people with anemia. It is misleading as it is currently. The current writing suggest that almost half of the childbearing women had mild anemia and the remaining half had moderate to sever anemia.
Response 3: We have revised the sentence in line 196-198.
Point 4: Table 1: Authors should indicate which groups were significantly different from each other based on post-hoc test results. For example, using superscripts, with groups with significantly different Hb concentration or weighted anemia prevalence values having different superscripts.
Response 4: We have compared the differences between subgroups in the revised manuscript.
Point 5: There is still need to improve the grammar and language usage in several parts of the manuscript. For example lines 304.
Response 5: We tried our best to improve the grammar and language usage in the revised manuscript including the sentence in Line 304.
Point 6: Also using “hemoglobin concentration” is to be preferred over “hemoglobin levels” as levels may also imply categories of hemoglobin concentration.
Response 6: We have revised it in the manuscript.
Reviewer 2 Report
Authors have answered all questions.
Thank you for adding information regarding the organization in China, this information is useful for western readers.
I´d advise to reprote total error in next studies, because it joins both constant and random errors
Author Response
Point 1: I´d advise to report total error in next studies, because it joins both constant and random errors
Response 1: Thank you for your suggestion. We will improve the statistics on errors in the following round of nutrition survey.